

# Combination of Vitamin C and Lenvatinib potentiates antitumor effects in hepatocellular carcinoma cells *in vitro*

Xinyue Wang[1,*], Songyi Qian[2,*], Siyi Wang[1], Sheng Jia[1], Nishang Zheng[1], Qing Yao[1] and Jian Gao[3]

[1] Department of Nutrition, Xiamen Clinical Research Center for Cancer Therapy, Zhongshan Hospital (Xiamen), Fudan University, Xiamen, Fujian province, China

[2] Department of Cardiac Surgery, Zhongshan Hospital (Xiamen), Fudan University, Xiamen, Fujian province, China

[3] Department of Nutrition, Zhongshan Hospital, Fudan University, Shanghai, China

[*] These authors contributed equally to this work.

## ABSTRACT

Lenvatinib has become a first-line drug in the treatment of advanced hepatocellular carcinoma (HCC). Investigating its use in combination with other agents is of great significance to improve the sensitivity and durable response of Lenvatinib in advanced HCC patients. Vitamin C (L-ascorbic acid, ascorbate, VC) is an important natural antioxidant, which has been reported to show suppressive effects in cancer treatment. Here, we investigated the effect of the combination of VC and Lenvatinib in HCC cells *in vitro*. We found that treatment of VC alone significantly inhibited the proliferation, migration and invasion in HCC cells. Additionally, VC was strongly synergistic with Lenvatinib in inhibition of the proliferative, migratory and invasive capacities of HCC cells *in vitro*. In conclusion, our results demonstrate that the combination of VC and Lenvatinib has synergistic antitumor activities against HCC cells, providing a promising therapeutic strategy to improve the prognosis of HCC patients.

## INTRODUCTION

Hepatocellular carcinoma (HCC) is the main histological type of primary liver cancer, which causes about 830,000 deaths every year (*Siegel, Miller & Jemal, 2020*). HCC is mainly caused by chronic infection with hepatitis B virus (HBV) or hepatitis C virus (HCV), alcohol abuse, or non-alcoholic fatty liver disease (NAFLD). Since symptoms of early HCC are usually not obvious, most patients are already in advanced stages at the time of diagnosis and lose the opportunity for local treatment such as radical hepatectomy, radiofrequency ablation, or transarterial therapy (*Abbasoglu, 2015*). Unlike other types of cancers including lung or breast cancer, advanced HCC has few therapeutic options because it is not sensitive to chemotherapy or radiotherapy. Receptor tyrosine kinase signaling pathways, such as the VEGF/VEGFR system, play an important role in regulating angiogenesis. VEGFR plays a central role as a sensor of angiogenesis, which is an important target of several tyrosine kinase inhibitors (TKIs) including Sorafenib and Lenvatinib. However, the efficacy and

Corresponding authors
Qing Yao,
yao.qing@zsxmhospital.com
Jian Gao, gao.jian@zs-hospital.sh.cn

duration of treatment of these drugs is still limited by acquired or inherent resistance. Therefore, identification of novel enhancers or synergists of TKIs has become an urgent need for clinical treatment of advanced HCC.

Lenvatinib has become a first-line drug in the treatment of advanced HCC. Lenvatinib is a multi-target inhibitor for VEGFR1-3, FGFR1-4, PDGFR $\alpha/\beta$, C-kit and RET, and exhibits strong antitumor activity (*Al-Salama, Syed & Scott, 2019*; *Hatanaka, Naganuma & Kakizaki, 2021*). The inhibition of FGFR4 by Lenvatinib is considered as a key factor in its antitumor effect (*Matsuki et al., 2018*). Lenvatinib inhibits both VEGF and FGF pathways and inactivates the proliferative signals in cancer cells, thereby inhibiting tumor growth and angiogenesis (*Tohyama et al., 2014*; *Yamamoto et al., 2014*). It can suppress the migration and invasion of cancer cells by regulating the expression of matrix metalloproteinases (MMPs) and tissue inhibitors of MMPs (TIMPs) (*He et al., 2018*). It also inactivates ERK, AKT and TGF-$\beta$ pathways, resulting in the suppression of cell migration, epithelial-mesenchymal transformation (EMT) and growth (*Zhang et al., 2020*). Nevertheless, a total of 85% of patients achieved a partial response or disease stabilization, many patients did not respond or acquired resistance, limiting the overall therapeutic efficacy of Lenvatinib (*Kudo et al., 2018*). For this reason, it is of great significance to improve the sensitivity and durable response of Lenvatinib in advanced HCC patients.

Vitamin C (L-ascorbic acid, ascorbate, VC) is an important natural antioxidant with a controversial history in cancer treatment. In the 1970s, Pauling and Cameron conducted clinical trials showing that intravenous injection of VC effectively prolong survival in patients with advanced cancer (*Cameron & Campbell, 1974*; *Cameron & Pauling, 1976*). However, the Mayo study found that oral VC did not show any benefit for tumors (*Creagan et al., 1979*; *Moertel et al., 1985*). It was later realized that the route of VC administration was the key reason for this difference, with intravenous injection of VC producing much higher plasma concentrations than oral VC (*Ngo et al., 2019*). A recent study reported that VC is able to kill cancer stem cells, and intravenous injection of VC has a significant effect on prolonging tumor-free survival of patients with HCC (*Lv et al., 2018*). However, whether VC enhances the anticancer effect of Lenvatinib remains unclear. In the present study, we aimed to evaluate the efficacy of combination of Lenvatinib and VC against HCC. We found that the combination of Lenvatinib and VC exhibits significant suppressive activity in HCC cells *in vitro*. Our study indicates that the combinational therapy of Lenvatinib and VC may be a promising strategy for HCC patients.

## MATERIAL AND METHODS

### Cell culture and drugs

Hep3B, Huh7 and SK-Hep-1 cells were obtained from the Fujian Provincial Key Laboratory of Chronic Liver Disease and Hepatocellular Carcinoma, Xiamen University. All cells were cultured in DMEM medium (Gibco) supplemented with 10% fetal bovine serum (FBS, Vivacell) at 37 °C in an atmosphere containing 5% $CO_2$. The authentication of these cell lines used in this present study was performed *via* comparisons with the STR database. VC and lenvatinib were purchased from Selleck Chemicals and MedChemExpress Company, respectively.

## Ethical statement

This study was approved by the Ethics Committee of Xiamen Branch, Zhongshan Hospital, Fudan University (ethics approval number: no. 2021–0910-DN).

## Colony formation assay

A total of 2,000 cells were seeded into six-well plates, treated with drug or drug vehicle and incubated for about 7–14 days until colonies were visible. Cell culture medium with different concentration of VC and/or lenvatinib were changed every three days during this period. Then, cells were fixed with 4% paraformaldehyde for 15 min and stained with 0.1% crystal violet solution for 30 min. The number of visible colonies (>50 cells/colony) were counted.

## Cell viability detection

Cell viability was assessed using cell counting kit-8 (CCK-8). A total of 2,000 cells were seeded into each well of 96-well plates in 100 μl culture medium and incubated overnight. On the following day, cells were treated with drug or drug vehicle. At indicated time point, 10 μl CCK-8 reagent was added to each well and incubated for another 1 h. The absorbance was measured at 450 nm with a multifunctional microplate reader (Thermo Fisher, Waltham, MA, USA).

## Transwell assay

Cells were pre-treated with drug or drug vehicle for 48 h. To detect cell migration, $1 \times 10^5$ cells were resuspended in serum-free DMEM and added into upper chambers in a transwell insert (Corning Inc., Corning, NY, USA). To detect cell invasion, $2 \times 10^5$ cells were resuspended in serum-free DMEM and added into upper chambers in a transwell insert coated with Matrigel (BD Biosciences, San Jose, CA, USA). A total of 500 μl DMEM medium supplemented with 10% FBS was added to the lower chamber. After incubation for 24 h, cells on the top of the membrane were wiped off with a cotton swab. Cells that migrated through the membrane were fixed with 4% paraformaldehyde for 15 min, stained with 0.1% crystal violet solution for 30 min and counted under a microscope.

## qRT-PCR assay

Total RNA from cells was extracted by using EasyPure® RNA Kit (Transgene, Beijing, China). A total of 1 μg RNA was reversely transcribed using TransScript® Uni All-in-One First-Strand cDNA Synthesis SuperMix for qPCR (One-Step gDNA Removal) (Transgene, Beijing, China). Then, qRT-PCR was performed in the Lightcycle Real-Time PCR System (Roche, Basel, Switzerland) using FastStart Universal SYBR Green Master (Rox) (Roche). The gene-specific primers are shown as follow: c-MET-forward: CCACGGGA-CAACACAATACA, c-MET-reverse: TAAAGTGCCACCAGCCATAG; ACTB-forward: GGAAATCGTGCGTGACATTAAG, ACTB-reverse: AGCTCGTAGCTCTTCTCCA. ACTB mRNA was taken as internal reference.

## Statistics

Data are presented as mean ± standard deviation (SD). All comparisons of means were calculated by using ANOVA tests with Tukey HSD correction for multiple means comparisons. A $P$ value less than 0.05 was considered statistically significant.

## RESULTS

### VC inhibits the proliferation, migration and invasion of HCC cells

First, we treated three different HCC cell lines with various concentration of VC, and the concentration-survival curves were plotted. The results showed that the 50% inhibitory concentration ($IC_{50}$) of VC in Hep3B and SK-Hep-1 cells was much lower than that in Huh7 cells (Fig. 1A), suggesting that Hep3B and SK-Hep-1 cells were more sensitive to VC than Huh7 cells. Treatment with VC suppressed HCC cell proliferation in a does- and time-dependent manner (Fig. 1B). Similarly, the VC treatment repressed the colony formation ability of HCC cells (Fig. 1C). However, VC did not show an obvious cytotoxic effect on THLE-2 cells, a normal liver cell line (Fig. S1).

The effect of VC on the migration and invasion was also measured. The results of transwell assays demonstrated that the migratory and invasive abilities of HCC cells was inhibited by the treatment of VC in a does-dependent manner (Figs. 2A and 2B).

### VC enhances the anti-proliferative activity of Lenvatinib in HCC cells

Next, we determined whether VC enhanced the effect of Lenvatinib on cell survival. we treated HCC cells with various concentrations of VC, Lenvatinib or a combination of both for 48 h. Treatment with Lenvatinib alone significantly inhibited cell survival, while cotreatment of VC and Lenvatinib had a greater inhibitory effect (Fig. 3A). Upon co-administration with VC, the $IC_{50}$ values of Lenvatinib were significantly decreased. Furthermore, cell proliferation was examined using CCK-8 and colony formation assays, and the results showed that the combination of VC and Lenvatinib exerts a more significant anti-proliferative effect on HCC cells (Figs. 3B–3C).

We then evaluated their synergism using the Chou-Talalay method with the Calcusyn software (Biosoft, Cambridge, UK). As shown in Table 1, a combination index (CI) less than 1 indicates the drug combination has synergism. The CI value can reach as low as 0.6883, demonstrating a moderate synergistic activity for this drug combination.

### VC reinforces the anti-metastatic effect of Lenvatinib in HCC cells

Although previous studies have reported that Lenvatinib can attenuated the tumor metastasis, whether its combination with VC enhanced this effect remains unknown. The transwell assays demonstrated that the cellular migration and invasion were significantly inhibited after treatment with the combination of VC and Lenvatinib compared with treatment with either of these drugs alone (Figs. 4A and 4B).

### VC suppresses the c-Met mRNA expression

Finally, we preliminarily investigated the underlying molecular mechanism. Previous study reported that VC represses the transcription of c-MET (*Chan, Yu & Yang, 1999*; *Lorenzato*

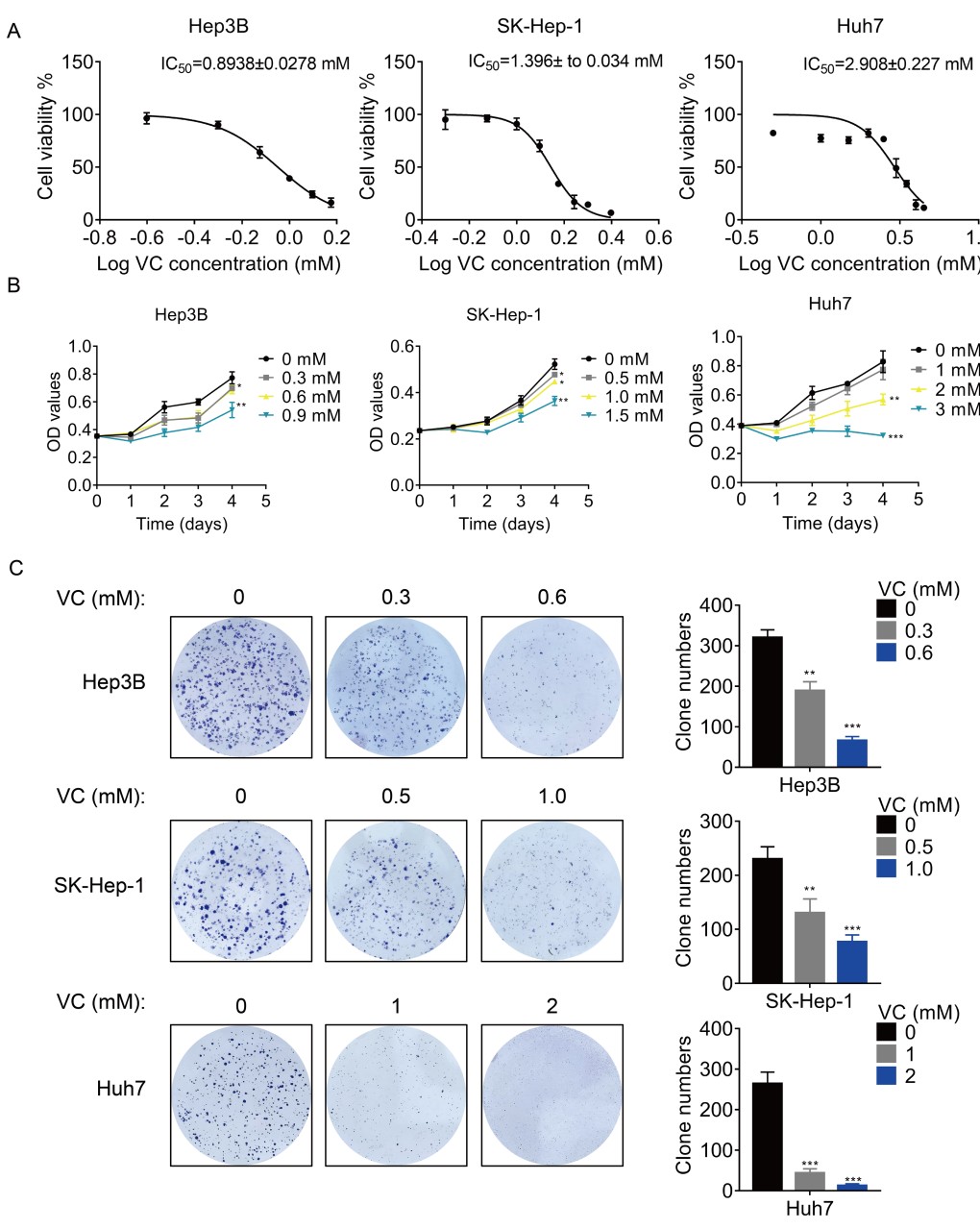

**Figure 1** **VC inhibits the proliferation, migration and invasion of HCC cells.** (A) Hep3B, SK-Hep-1 and Huh7 cells were treated with various concentrations of VC for 48 h. IC50 value of VC in HCC cell lines was then determined. (B) Hep3B, SK-Hep-1 and Huh7 cells were treated with various concentrations of VC. At different time points, the cell proliferation was determined by CCK-8 assay. (C) The clone formation of Hep3B, SK-Hep-1 and Huh7 cells treated with various concentrations of VC were detected. The representative images and statistical results were shown. * $p < 0.05$, ** $p < 0.01$, *** $p < 0.001$.

*et al., 2020*; *Pathi et al., 2011*; *Ye et al., 2019*). C-MET has been found to be important for the resistance of Lenvatinib (*Fu et al., 2020b*; *Nakagawa et al., 2014*). Therefore, we detected the effect of VC treatment on c-MET mRNA expression. The results of qRT-PCR showed

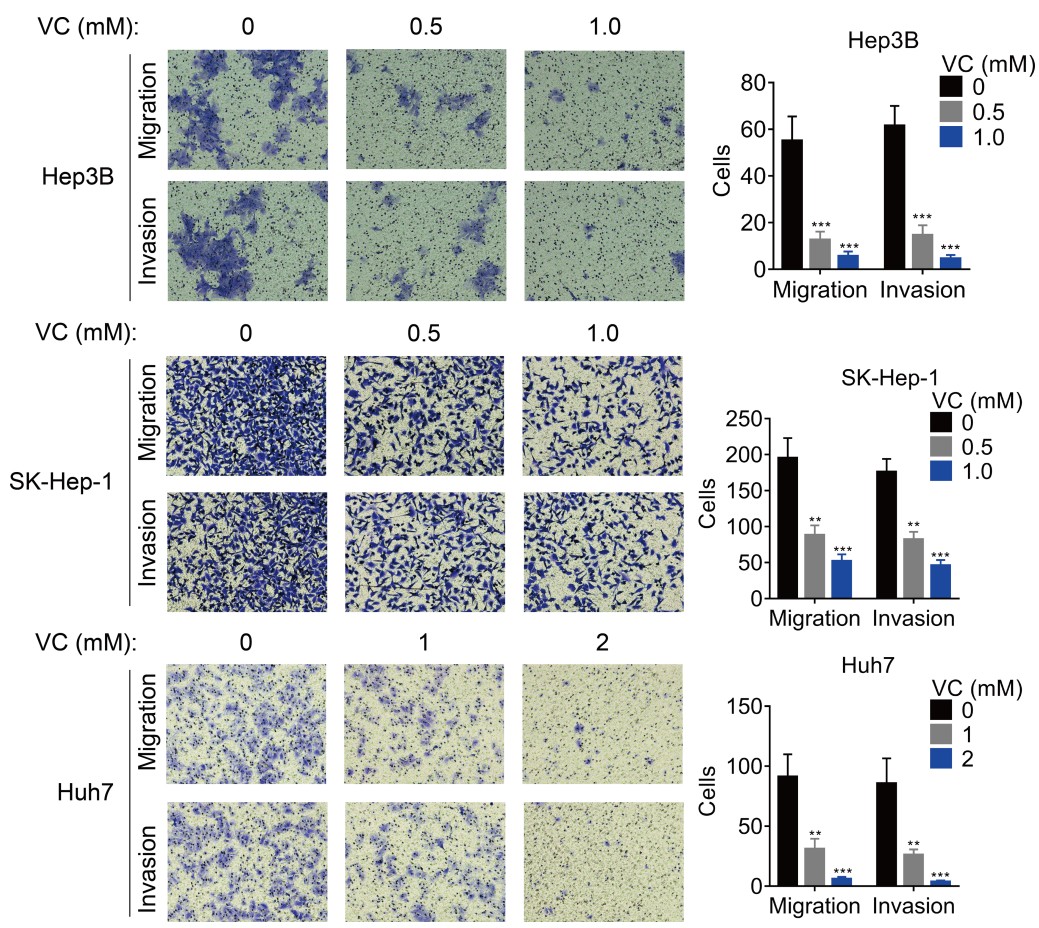

**Figure 2** **VC inhibits the proliferation, migration and invasion of HCC cells.** Hep3B, SK-Hep-1 and Huh7 cells were treated with various concentrations of VC for 48 hr. A transwell assay was then conducted to examine the migratory and invasive capabilities of the HCC cells. The representative images and statistical results of the transwell assays were shown. ** $p < 0.01$, *** $p < 0.001$.

that administration of VC significantly downregulated the mRNA expression of c-MET (Fig. 5).

## DISCUSSION

The inhibition of one canonical signaling will activate another compensatory pathway in the treatment of cancer, leading to the resistance against drugs. Therefore, combination therapies applying agents aiming at different targets is prevalent in cancer managements. In 2018, Lenvatinib was approved by the FDA as a first-line treatment for advanced patients with HCC. However, some patients with advanced HCC who received Lenvatinib therapy acquired drug resistance after a period of time. Recently, the underlying mechanism of Lenvatinib therapy in HCC has been revealed. In specific, inhibition of fibroblast growth factor receptor (FGFR) by Lenvatinib treatment leads to feedback activation of the EGFR-PAK2-ERK5 signaling axis. The combination of the EGFR inhibitor Gefitinib or Erlotinib

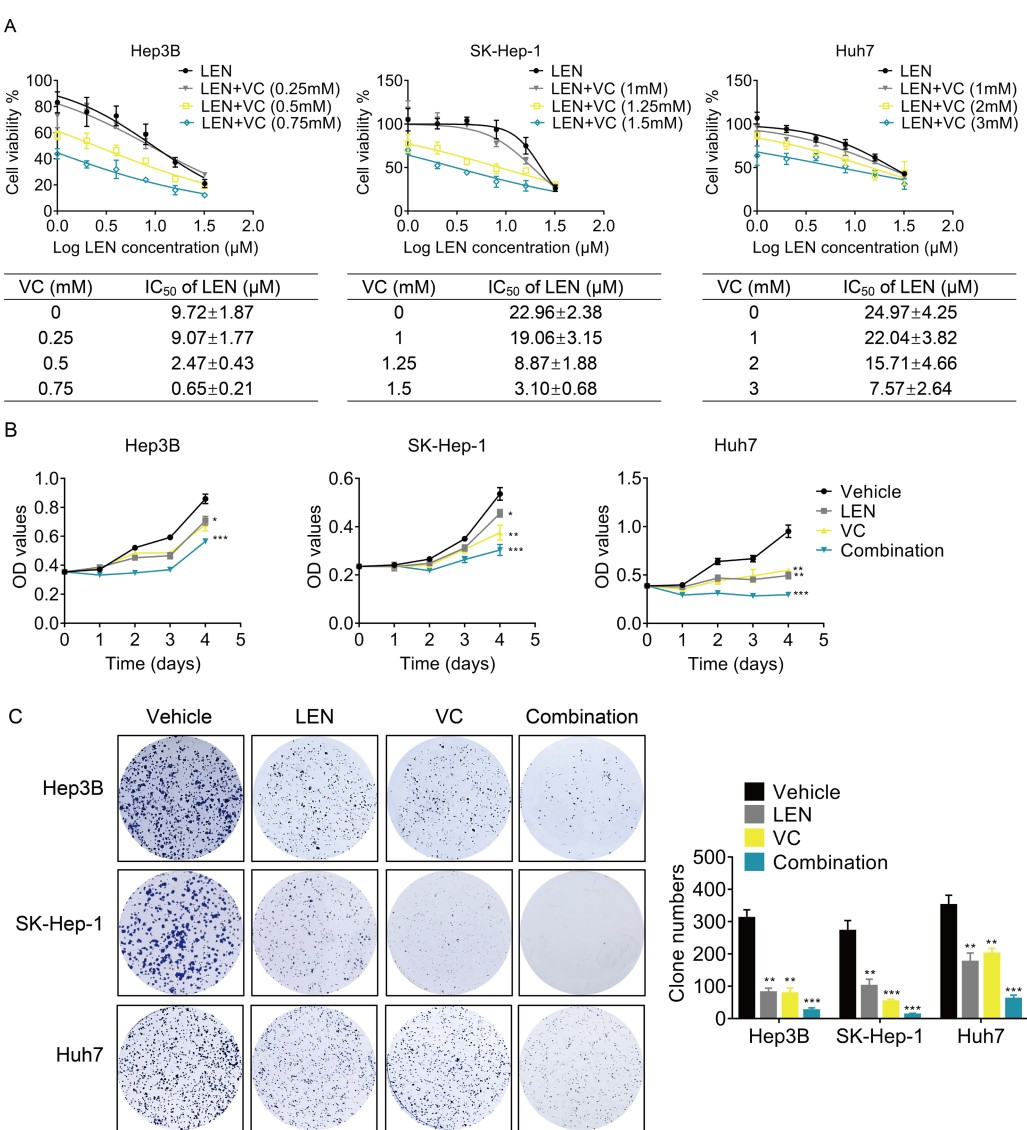

**Figure 3** **VC enhances the anti-proliferative activity of Lenvatinib in HCC cells.** (A) Hep3B, SK-Hep-1 and Huh7 cells were treated with various concentrations of VC, Lenvatinib, or both of the drugs for 48 h. IC50 value of VC in HCC cell lines was then calculated. (B) Hep3B, SK-Hep-1 and Huh7 cells were treated with VC and Lenvatinib individually or in combination. At different time points, the cell proliferation was determined by CCK-8 assay. (C) The clone formation of Hep3B, SK-Hep-1 and Huh7 cells treated with VC and Lenvatinib individually or in combination were detected. The representative images and statistical results were shown.$^*$ $p < 0.05$, $^{**}$ $p < 0.01$, $^{***}$ $p < 0.001$.

and Lenvatinib displays enhanced anti-proliferative effects (*He et al., 2022*; *Jin et al., 2021*). HGF/c-MET signaling is closely associated with drug resistance in cancer. In HCC cells with high c-MET expression, HGF reduces the antiproliferative and anti-invasive effects of Lenvatinib. Using the c-MET inhibitor PHA-665752 or Golvatinib is capable to abolish HGF-induced Lenvatinib resistance (*Fu et al., 2020b*; *Nakagawa et al., 2014*). Moreover, using CRISPR/Cas9 library screening method, two key resistance genes, neurofibromin

1(NF1) and dual specificity phosphatase 9 (DUSP9), are screened out as critical drivers for Lenvatinib resistance in HCC cells. Trametinib, an inhibitor of MEK pathway, can be used to reverse Lenvatinib resistance mediated by NF1 and DUSP9 loss in HCC cells (*Lu et al., 2021*). These findings suggest that targeting the resistance-associated molecules or pathway will improve the anti-tumor effect of Lenvatinib therapy for HCC.

VC is transported into cells primarily through sodium-dependent vitamin C-specific transporters (SVCTs). For liver, SVCT-2 is a key protein responsible for VC uptake, and the inhibitory effect of VC on HCC cells is related to the expression of SVCT-2 (*Lv et al., 2018*). Here, we found that Hep3B and SK-Hep-1 cells were more sensitive to VC treatment than Huh7 cells, which may due to the differential expression levels of SVCT-2. Magnesium supplementation can activate SVCT-2 and increase the Vmax value of SVCT-2, which show synergistic anticancer effects with VC in cancer cells with low expression level of SVCT-2 (*Cho et al., 2020*). Thus, cotreatment with $MgSO_4$ or $MgCl_2$ solution may improve the sensitivity of VC in Huh7 cells.

VC increases ROS levels in tumor cells, causing DNA damage and ATP depletion, leading to cell cycle arrest and apoptosis (*Lv et al., 2018*). In addition, VC plays an important role in the composition, structure, and biomechanical characteristics of the extracellular matrix by regulating the amount and molecular structure of collagen, and suppresses hypoxia-inducible factor (HIF), thereby inhibiting tumor metastasis and angiogenesis (*Fu et al., 2020a*). Consistently, we found that VC significantly inhibited the proliferation, migration and invasion of HCC cells. Notably, VC inactivates PAK2, represses the transcription of c-Met and inhibits the EGFR-MAPK/ERK signaling (*Chan, Yu & Yang, 1999*; *Lorenzato et al., 2020*; *Pashirzad et al., 2021*; *Pathi et al., 2011*; *Ye et al., 2019*). These important molecules and signaling pathways are already demonstrated to be closely associated with Lenvatinib resistance and metastasis in cancer cells (*Fu et al., 2020b*; *Jin et al., 2021*; *Nakagawa et al., 2014*; *Shao et al., 2020*). Our results further confirmed the suppressive effect of VC on c-MET mRNA expression in HCC cells. Altogether, these findings suggested that VC may improve the resistance of Lenvatinib therapy and repress metastasis in HCC *via* regulating c-MET. On the other hand, our results of this study showed that treatment with the combination of VC and Lenvatinib showed a better anti-proliferative and anti-metastatic effect compared with treatment with either of these drugs alone. However, the *in vivo* experiments and exact mechanism need further investigation.

Recent studies have found that Lenvatinib combined with anti-PD-1 therapy plays a unique immunomodulatory role by activating immune pathways, reducing Treg cell infiltration and inhibiting TGF$\beta$ signaling (*Torrens et al., 2021*). In addition, Lenvatinib reduces PD-L1 level of HCC cells and Treg differentiation by blocking FGFR4, which improves anti-PD-1 efficacy. Lenvatinib enhances the proteasome degradation process of PD-L1 in HCC cells by blocking FGFR4-GSK3$\beta$ pathway. The sensitivity of IFN-$\gamma$-pretreated HCC cells to the killing effect of T cell can be restored by targeting FGFR4. On the other hand, interleukin-2 (IL-2) levels increased after anti-PD-1 treatment, but Lenvatinib blocked IL-2-mediated Treg differentiation by targeting FGFR4 and inhibiting signal transduction and STAT5 phosphorylation (*Yi et al., 2021*). Interestingly, VC also

**Table 1  The combination index calculated by CalcuSyn software.**

|  | Lenvatinib ($\mu$M) | Vitamin C (mM) | CI |
|---|---|---|---|
| Hep3B | 4 | 0.25 | 1.4424 |
|  | 8 | 0.5 | 0.8928 |
|  | 16 | 0.75 | 0.7166 |
| SK-Hep-1 | 4 | 1 | 1.3114 |
|  | 8 | 1.25 | 0.7984 |
|  | 16 | 1.5 | 0.6495 |
| Huh7 | 4 | 1 | 1.2010 |
|  | 8 | 2 | 0.8321 |
|  | 16 | 3 | 0.6883 |

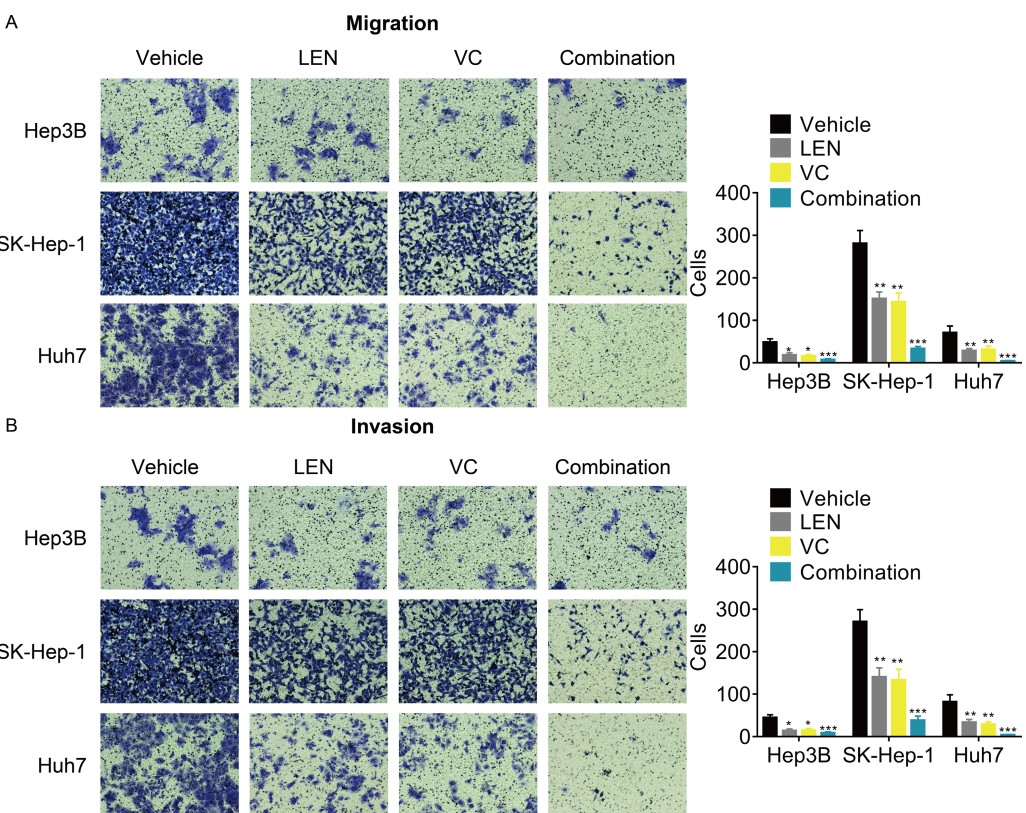

**Figure 4  VC reinforces the anti-metastatic effect of Lenvatinib in HCC cells.** Hep3B, SK-Hep-1 and Huh7 cells were treated with VC and Lenvatinib individually or in combination for 48 hr. A transwell assay was then conducted to examine the migration (A) and invasion (B) of the HCC cells. The representative images and statistical results of the transwell assays were shown. $*$ $p < 0.05$, $**$ $p < 0.01$, $***$ $p < 0.001$.

enhances the effect of anti-PD-1 immunotherapy, which may be achieved by inhibiting IL-6-mediated pD-L1 protein stability (*Chan et al., 2019*; *Luchtel et al., 2020*). These findings suggest that combination of VC and Lenvatinib may augment antitumor immune response of anti-PD-1 in HCC.

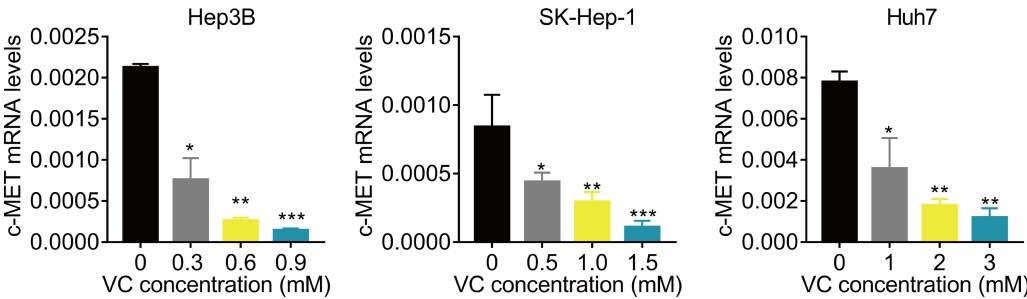

**Figure 5  VC suppresses the c-Met mRNA expression.** Hep3B, SK-Hep-1 and Huh7 cells were treated with various concentrations of VC. 48 h later, the c-MET mRNA levels were measured using qRT-PCR assay. $*$ $p < 0.05$, $**$ $p < 0.01$, $***$ $p < 0.001$.

In sum, our current study implicated that the cotreatment of VC and Lenvatinib can improve treatment efficacy. Nonetheless, further *in vivo* study, as well as clinical evidences are still needed to confirm this clinical translation.

## ACKNOWLEDGEMENTS

We thank the Fujian Provincial Key Laboratory of Chronic Liver Disease and Hepatocellular Carcinoma, Xiamen University for providing the HCC cell lines.

### Funding

This research is supported by the Incubation Fund of Zhongshan Hospital, Fudan University (Xiamen Branch) (No. 2020ZSXMYS24). The funders had no role in study design, data collection and analysis, decision to publish, or preparation of the manuscript.

### Grant Disclosures

The following grant information was disclosed by the authors:
The Incubation Fund of Zhongshan Hospital, Fudan University (Xiamen Branch): 2020ZSXMYS24.

### Competing Interests

The authors declare there are no competing interests.

### Author Contributions

- Xinyue Wang conceived and designed the experiments, performed the experiments, analyzed the data, prepared figures and/or tables, authored or reviewed drafts of the article, and approved the final draft.
- Songyi Qian conceived and designed the experiments, performed the experiments, analyzed the data, prepared figures and/or tables, authored or reviewed drafts of the article, and approved the final draft.

- Siyi Wang performed the experiments, analyzed the data, authored or reviewed drafts of the article, and approved the final draft.
- Sheng Jia performed the experiments, analyzed the data, authored or reviewed drafts of the article, and approved the final draft.
- Nishang Zheng performed the experiments, analyzed the data, authored or reviewed drafts of the article, and approved the final draft.
- Qing Yao conceived and designed the experiments, performed the experiments, analyzed the data, prepared figures and/or tables, and approved the final draft.
- Jian Gao conceived and designed the experiments, performed the experiments, analyzed the data, prepared figures and/or tables, and approved the final draft.

## Data Availability

The raw data are available in the Supplemental Files.

## Supplemental Information

Supplemental information for this article can be found online at http://dx.doi.org/10.7717/peerj.14610#supplemental-information.

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
