# Peer review of "Combination of Vitamin C and Lenvatinib potentiates antitumor effects in hepatocellular carcinoma cells *in vitro"

_PeerJ, doi:10.7717/peerj.14610_

## Round 0.1 · original submission · Major Revisions

The reviewers have raised significant concerns about your work. I am giving you an opportunity to address these, but this will require extra experimental work and careful rebuttal of all points. Without this, we cannot proceed.

You need to address the issue of how you score and present your results (reviewer-2). This raises concerns about interpretation. Please address this point fully and extensively and provide the requested analysis.

The cells chosen are also problematical. You should address this with other examples of HCC lines, and provide clearly shown negative controls of non-HCC lines. The issue of non-specific toxicity must also be addressed with further work.

Proof reading is required throughout, and all primary data (all technical and biological replicates) must be presented as supplemental data.

Reviewer 1 ·

Basic reporting

This study investigated the effect of combination lenvatinib plus vitamin C in the proliferation, migration and invasion in three hepatocellular carcinoma cell lines. Though the study is interesting, some concerns which would improve the quality of the manuscript should be addressed.

Experimental design

What’s the specific dosage and method of use about lenvatinib and vitamin C?

Validity of the findings

Only basic experiment with three cell lines was performed. Experiment in vivo should be performed to confirm their findings. And the mechanisms of the effect of the combination treatment is also unknown.

Additional comments

1. The format of the references is not uniform in Background.
2. Refs about vitamin C should be provided in the front two sentences (the third paragraph of Background).

Reviewer 2 ·

Basic reporting

The author never discussed about Figure 4. This appears to be necessary to reinforce the claim of synergistic effect on migrastatic effect of the combo regimen. Without this supportive, and proven statement, the claim 'our results demonstrate that the combination of VC and Lenvatinib has
31 synergistic antitumor activities against HCC cells, providing a promising therapeutic strategy to
32 improve the prognosis of HCC patients' cannot be testified.

The manuscript was written using professional English with partial editing suggestions. Relevant references were used, help audience to understand the background/context of the problem.
- Please check citation style in introduction, first paragraph, line 36, 40

Experimental design

1. The authors attempted to assess the synergistic effect between VC and lenvatinib. To accurately score synergistic effects, the authors are advised to report their findings using acceptable scoring methodologies such as the Bliss or Loewe methods. Current data in Fig 3, simply showed using IC50 scoring only. Readers will not be able to distinguish the effects, either only additive or truly synergistic.

2. The proposed experiment in evaluating proliferation, migration and invasion assays with 3 HCC cell lines is a good starting point to assess the tentative synergistic effects. However, the study lacks strong negative control. How do we know that the observed effects will be specific to cancer cells only, and not due to toxicity effect? This may require comparison with normal cells.

Validity of the findings

1. The most critical issue in this study is whether the claim synergistic effect is truly the case. One cannot assess this without proper methodology (Bliss, Loewe scores as suggested).
2. The limited number of HCC cells (Hep3B, Huh7 is in the same subgroup CL1 (https://doi.org/10.1053/j.gastro.2019.05.001) and SK-HEP1 has been shown to be contaminated with endothelial origin (10.1016/j.jhep.2017.08.002)) posts another important caution for the ability to generalize results from this study, to the proposed claim, in identifying combo regimens for HCC patients.
3. As stated earlier, the author did not discuss their finding on 1 full figure (figure 4). This posts another important issue on the validity of the proposed claims.

Additional comments

The manuscript in general quickly jumped to the conclusion that Lenvatinib+VC showed synergistic effects without the necessary synergistic scoring. The overstated translational power of these results, as a novel combo regimen for HCC, is not logical with the provided dataset.

---

## Round 0.2 · Minor Revisions

Thanks for the excellent way in which you have addressed the concerns of the reviewers. One has commented that they think you are still 'over-reaching' in the conclusions, and I am minded to agree. Could you tone this down and re-submit please?
Thanks

Reviewer 1 ·

Basic reporting

good work

Experimental design

no problem

Validity of the findings

novel finding

Additional comments

none

Reviewer 2 ·

Basic reporting

Improved writing with appropriate citations. Authors also added the requested citations.

Experimental design

Authors added the requested appropriate methodologies (and associated results) to score synergistic effect.

Validity of the findings

My only concern left is on the still overclaimed statement, "In sum, our current study suggests that cotreatment of VC and Lenvatinib achieved a superior anti-HCC effect compared with treatment with either of these drugs alone, and VC has great potential to be used as a synergistic agent of Lenvatinib to treat HCC patients." I would suggest slightly smaller claim, e.g. "implicating that the cotreatment of VC and Lenvatinib can improve treatment efficacy. Nonetheless, further in vivo study, as well as clinical evidences are still needed to confirm this clinical translation".

Additional comments

The authors have worked carefully to addressed the points raised in prior round of revision. Apart from revising the overclaimed statement on clinical translation, I have no further comments.

---

## Round 0.3 · accepted · Accept

Thanks for attending to the remaining issue.